# Mannan-Binding Lectin Is Associated with Inflammation and Kidney Damage in a Mouse Model of Type 2 Diabetes

**DOI:** 10.3390/ijms25137204

**Published:** 2024-06-29

**Authors:** Gry H. Dørflinger, Charlotte B. Holt, Steffen Thiel, Jesper N. Bech, Jakob A. Østergaard, Mette Bjerre

**Affiliations:** 1Medical/Steno Aarhus Research Laboratory, Department of Clinical Medicine, Aarhus University, 8200 Aarhus, Denmark; grydoe@clin.au.dk (G.H.D.); charlotte.brinck.holt@clin.au.dk (C.B.H.); 2Department of Internal Medicine, Regional Hospital Gødstrup, 7400 Herning, Denmark; jesper.noergaard.bech@goedstrup.rm.dk; 3Department of Endocrinology and Internal Medicine, Aarhus University Hospital, 8200 Aarhus, Denmark; jakob.oestergaard@clin.au.dk; 4Department of Biomedicine, Aarhus University, 8200 Aarhus, Denmark; st@biomed.au.dk; 5University Clinic in Nephrology and Hypertension, Department of Clinical Medicine, Aarhus University, 8200 Aarhus, Denmark; 6Steno Diabetes Center Aarhus, 8200 Aarhus, Denmark

**Keywords:** diabetic nephropathy, MBL, complement, inflammation, OB, BTBR mice

## Abstract

Autoreactivity of the complement system may escalate the development of diabetic nephropathy. We used the BTBR OB mouse model of type 2 diabetes to investigate the role of the complement factor mannan-binding lectin (MBL) in diabetic nephropathy. Female BTBR OB mice (*n* = 30) and BTBR non-diabetic WT mice (*n* = 30) were included. Plasma samples (weeks 12 and 21) and urine samples (week 19) were analyzed for MBL, C3, C3-fragments, SAA3, and markers for renal function. Renal tissue sections were analyzed for fibrosis, inflammation, and complement deposition. The renal cortex was analyzed for gene expression (complement, inflammation, and fibrosis), and isolated glomerular cells were investigated for MBL protein. Human vascular endothelial cells cultured under normo- and hyperglycemic conditions were analyzed by flow cytometry. We found that the OB mice had elevated plasma and urine concentrations of MBL-C (*p* < 0.0001 and *p* < 0.001, respectively) and higher plasma C3 levels (*p* < 0.001) compared to WT mice. Renal cryosections from OB mice showed increased MBL-C and C4 deposition in the glomeruli and increased macrophage infiltration (*p* = 0.002). Isolated glomeruli revealed significantly higher MBL protein levels (*p* < 0.001) compared to the OB and WT mice, and no renal MBL expression was detected. We report that chronic inflammation plays an important role in the development of DN through the binding of MBL to hyperglycemia-exposed renal cells.

## 1. Introduction

Diabetic nephropathy affects one out of three patients with diabetes and is the leading cause of renal failure requiring dialysis or kidney transplantation in the Western world [1,2]. An early event following chronic hyperglycemia is endothelial dysfunction, which is associated with the progression of diabetic nephropathy [3]. Various cells in the kidneys, including glomerular and vascular endothelial cells, have been associated with the development of diabetic nephropathy [4]. 

The complement system is a central part of the innate immune system and plays a vital role in the first line of defense. Several studies have linked the complement system to the development of diabetic microvascular complications [5,6,7,8]. 

The complement system is activated through either of three distinct pathways: the classical pathway, the alternative pathway, or the lectin pathway. The lectin pathway of complement is activated when binding to a surface occurs by pattern-recognition molecules, including the two carbohydrate-recognizing collectins, mannan-binding lectin (MBL) and collectin-LK, as well as the three ficolins (H-, L-, and M-ficolin) [9,10]. Upon binding of, e.g., MBL to its target, MBL-associated serine proteases (MASPs) initiate downstream complement activation. Several complement factors are cleaved during this process, resulting in the formation of anaphylatoxins, C3a and C5a, as well as the deposition of opsonic factors C3b and iC3b on cell membranes [11]. In addition, complement activation initiates the formation of the membrane attack complex (MAC) that disrupts the cell membrane [11]. 

Autoreactivity of the complement system in diabetes may be initiated by hyperglycemia-induced glycation of surface amino acids, forming non-enzymatic advanced glycation end-products (AGEs) [12]. Patients with type 1 diabetes (T1D) have higher serum MBL levels than healthy individuals [13], and high MBL levels have been associated with microvascular complications [6,14]. Furthermore, MBL genotypes translating to high MBL levels are associated with increased mortality among patients with T1D during 12 years of follow-up [15]. We have recently shown that T2D patients with the high expression MBL genotype presented with impaired renal function [16]. Our previous studies have shown that MBL accumulates in the kidneys of diabetic mice (T1D) but not in the non-diabetic control mice, which suggests that diabetes induces carbohydrate patterns that are recognized by MBL [17,18,19]. Here, we investigate if this is also the case in T2D and potentially corresponds to the activation of the complement system. We use a robust animal model for type 2 diabetes and nephropathy (BTBR. Cg-Lep^ob^/WiscJ; Strain #004824) (OB) with progressive proteinuria and glomerular histopathology closely resembling human diabetic nephropathy [20,21]. It is an animal model that exhibits obesity and hyperphagia, develops profound insulin resistance and hyperinsulinemia, and progresses to diabetes. As the model exhibits hypotension [21], the factors related to hypertension in the pathophysiology of nephropathy are bypassed. A better understanding of the mechanisms for the development of diabetic nephropathy is needed to pursue novel therapeutics targeting diabetic nephropathy. 

## 2. Results

### 2.1. Diabetes Status, Kidney Weight, and Kidney-to-Body Weight Ratio

The OB mice presented a diabetic phenotype with higher HbA1c than the non-diabetic wild-type mice at week 12 and at study termination (Figure 1A). The OB mice had higher blood glucose levels than the WT mice (Figure 1B). The development in blood glucose levels during the study period, compared to the area under the curve (AUC) (Figure 1C), showed that OB mice had a statistically significant higher glucose level than the WT mice. As expected, the OB mice rapidly increased in body weight (Figure 1D), leading to a profound difference between the groups at the end of the study (*p* < 0.0001) (Figure 1E). The kidney weight was significantly higher in the OB group (Figure 1F). However, when normalizing kidney weight to body weight, no difference was observed (Figure 1G). 

### 2.2. Albumin-to-Creatinine Ratio and Plasma Cystatin C

We found a significantly higher urine albumin-to-creatinine ratio (UACR) in the OB mice as compared to the non-diabetic WT mice, *p* < 0.0001 (Figure 2A). These findings are consistent with the assessment of kidney function by plasma cystatin C at week 21, showing a tendency toward a higher plasma cystatin C in the OB group (Figure 2B); however, they are not statistically significant (*p* = 0.07). The plasma cystatin C levels decreased over time in the WT mice, whereas the levels increased in the OB mice (Figure 2B). To evaluate if the difference in cystatin C was already evident in the early stages of this animal model, we measured plasma cystatin C in 5-week-old mice, but no difference between OB and WT mice was observed (Appendix A).

### 2.3. Immunohistochemistry

Renal changes in diabetes were subsequently evaluated by renal morphology and immunohistochemistry (Figure 3A). The glomerular area was used as a measure of glomerular hypertrophy and showed a significant increase in glomerular area in the OB group compared to WT (*p* < 0.0001) (Figure 3C). Furthermore, a significantly increased mesangial area fraction was observed in OB mice as compared to WT mice (*p* < 0.001) (Figure 3B).

### 2.4. Fibrosis

Morphological alterations towards fibrosis were assessed by the deposition of fibronectin and collagen in the glomeruli. Compared with the WT mice, we found a statistically significant increase in the deposition of fibronectin in the glomerular area of the OB mice, *p* < 0.0001 (Figure 3D). In comparison, no difference was observed in collagen deposition (data not shown).

To assess the impact of diabetes on the expression of mRNA from genes related to fibrosis, we analyzed the expression of mRNA encoding collagen 1, 3, and 4 (Col1a1, Col3a1, Col4a1), fibronectin (Fn1), matrix metalloproteinases-2 and -9 (Mmp2 and Mmp9), tissue inhibitors of metalloproteinases-1 and -2 (Timp1 and Timp2), as well as the mediator of fibrogenesis TGF-β (Tgfb1) and the mediator of the TGF-β action connective tissue growth factor (Ctgf) (Figure 4A). We found a down-regulation of the expression of the collagen genes and the fibronectin gene in the OB mice when compared to the WT mice. The expression of mRNA encoding MMP2 and TIMP2 was decreased in the OB mice, whereas expression from the genes encoding MMP9, TIMP-1, TGF-β, and CTGF showed no significant difference when compared to the WT mice (Figure 4A). 

### 2.5. The Complement System and Inflammation

We found significantly higher MBL-A plasma levels in the OB mice at 12 weeks, *p* < 0.0001, whereas no difference was found at week 21, *p* = 0.74. The concentration of MBL-A was almost twice as high in OB mice at week 5 when compared to WT mice, *p* < 0.001 (Appendix A). The MBL-A levels in the WT mice did not change over time (*p* = 0.2) (Figure 5A). Similarly, plasma MBL-C levels were significantly higher in the OB group at 12 weeks and 21 weeks as compared to WT, *p* < 0.0001 (Figure 5B), and decreased over time in the OB group, *p* = 0.003, but not in the WT, *p* = 0.18. As for the MBL-A, the plasma MBL-C was twice the concentration of the 5-week-old WT mice, *p* = 0.007 (Appendix A). We investigated the excretion of MBL-C in urine and found that the concentration was more than doubled in the OB compared with the WT mice, *p* < 0.001 (Figure 5E).

No difference in total C3 was detected at 5 weeks or 12 weeks (Appendix A, Figure 5C), but we found significantly higher levels of plasma C3 levels in the OB mice at 21 weeks (*p* < 0.001), although when comparing total C3 at week 12 (WT: 1295; IQR = 214 and OB: 1233; IQR = 406) and week 21 (WT: 761; IQR = 135 and OB: 990; IQR = 449), the level was significantly decreased at week 21 in both WT (*p* < 0.0001) and the OB mice (*p* < 0.001) (Figure 5). These data are in line with the gene expression analysis showing a higher expression of C3 mRNA in the OB mice (Figure 4B). A decrease in C3 levels from 12 to 21 weeks was observed in both the WT group and the OB group (*p* = 0.01). The level of complement C3 fragments C3d and iC3b was quantified as a measure of complement activation. We found no statistical difference in the levels of C3 fragments between the WT and the OB groups, neither in the 5-week-old mice (Appendix A) nor in the 21-week-old mice (Figure 5D), and no difference was found in the C3 fragment/total C3 ratio (data not shown). 

We analyzed the expression of selected genes with relevance for the activation of the complement system and inflammation in the kidney tissue. In addition to increased C3 mRNA in OB mice, we found an increased expression of mRNA encoding TNF-α in the OB mice as compared to the WT mice (*p* = 0.014). The NLRP3 inflammasome caspase 1 (Casp1) had a significantly lower expression in the OB mice compared to the WT mice, *p* = 0.006 (Figure 4B). We analyzed Nlrp3 gene expression as this inflammasome is required for caspase-1 activation and expression of the apoptosis-associated speck-like protein containing a CARD (ASC) gene expression due to the interaction for inflammasome formation. However, neither Nlrp3 nor Acs gene expression was significantly altered in the diabetic mice (Figure 4B).

The general inflammation was evaluated by measuring serum amyloid A 3 (SAA-3) in plasma. The OB mice were presented with significantly increased SAA-3 levels (*p* < 0.0001) (Figure 6), and this was already present at 5 weeks (*p* = 0.002) (Appendix A). The level of SAA3 decreased significantly from 12 to 21 weeks in both the WT mice (*p* < 0.0001) and the OB mice (*p* = 0.021). No correlations were found between SAA-3 and either MBL-A or MBL-C. 

### 2.6. Complement Staining and Inflammation in Glomeruli

We found evidence of increased MBL-binding and activation of the complement system in the glomeruli when investigating the immunofluorescence staining for MBL-C and C4 (Figure 7A). The staining for MBL-C was more intense in the OB mice compared to the staining in the WT mice (Figure 7B), *p* = 0.0003. We subsequently stained for C4 deposition (in the form of C4b) as an indication of complement activation through the lectin and/or the classical pathway. We found an increased fluorescence intensity of C4 in the OB mice compared to the WT, *p* < 0.0001 (Figure 7C). Furthermore, we observed that C4 staining was primarily localized in the juxta-glomerular area in the OB group as compared to the WT, in which the staining was evenly distributed throughout the glomerular area (Figure 7D). 

The localization of C4 staining in the glomeruli was evaluated by podocyte staining using nephrin (Figure 8A). The increased C4 deposition in OB mice did not seem to be situated in the proximity of the podocytes. However, staining by nNOS, visualizing the tubular region and the macula densa region, showed a potential involvement of the mesangial area, close to the macula densa region and the nearby situated tubular structure (Figure 8B).

We subsequently analyzed the homogenized snap-frozen kidney cortex and the snap-frozen, strainer-isolated glomeruli for the presence of MBL-A and MBL-C proteins. We found a significantly increased accumulation of MBL-A and MBL-C in both the kidney cortex and glomeruli (Figure 9). We further analyzed renal MBL expression to investigate if the detected MBL-A and MBL-C originated from local expression. However, in the gene expression analysis, mbl1 (encoding MBL-A) and mbl2 (MBL-C) Ct levels were above 32 in both groups, indicating a very low expression and rendering statistical analysis irrelevant. Thus, we conclude that no local MBL-A or MBL-C production at a relevant level was present in the kidney tissue and that the detected MBL originated from the deposition of circulating MBL. 

Detection of glomerular macrophage infiltration by CD68 staining showed a significantly increased count of CD68+ cells in the glomeruli of the OB mice as compared to WT mice, supporting increased local inflammation in the OB mice, *p* < 0.001 (Figure 3E).

### 2.7. MBL Adhesion In Vitro

In an in vitro experiment, human vascular endothelial cells were cultured in either hyperglycemic (21.5 mM glucose) or normoglycemic (5.5 mM glucose) conditions for 15 days (+/−5 days). In line with our in vivo findings indicating MBL autoreactivity in diabetes, we found a higher binding of MBL to HUVECs when cultured in a hyperglycemic environment compared to a normoglycemic environment, *p* = 0.03 (Figure 10).

## 3. Discussion

The present results support our hypothesis that the autoreactivity of MBL towards kidney tissue is increased in a mouse model of type 2 diabetes and nephropathy. We found increased activation of the complement system by measuring MBL and C4b staining in glomeruli, but no signs of local transcription of MBL in the kidney cortex. Furthermore, we showed that a hyperglycemic environment increased the adhesion of MBL. This supports the theory that prolonged hyperglycemia in diabetes can lead to glycoprotein alterations, which may enable adverse complement activation through the binding of MBL to neoepitopes.

Diabetic nephropathy includes glomerular mesangial expansion, podocyte loss, basement membrane thickening, and endothelial cell destruction [22]. Furthermore, a combination of hyperglycemia, oxidative stress, elevated lipid levels, the presentation of damage-associated molecular patterns (DAMPs), and advanced glycation end-products (AGEs) triggers pro-inflammatory signaling pathways, leading to the recruitment of macrophages [23]. Our findings in the OB mouse model of T2D and nephropathy support that inflammation, initiated via the recognition of altered self-cells by MBL, plays a crucial role in the development of diabetic nephropathy, likely through hyperglycemic changes and DAMPs. The OB mice presented with increased blood glucose, structural renal changes, increased inflammation both locally and in the circulation, and increased renal infiltration of CD68-positive cells, likely with MBL functioning as an opsonin. 

Traditionally, diabetic nephropathy has not been considered an immune-mediated disease, but recently, several indications of the involvement of the immune system have been discovered [5,24]. We previously showed that circulating MBL-C levels increase in C57Bl6 female mice when diabetes was induced by STZ injections mimicking T1D [25]. Supporting this, in the OB mice from the present study, the plasma levels of MBL-A and MBL-C were already 2-fold higher in the 5-week-old mice. In analogy, higher plasma levels of MBL are observed in children as compared to adults [26,27]. We found decreased levels of MBL-A and MBL-C from week 12 to week 21 in the OB mice. This observation fits with an inverse correlation between circulating MBL concentration and obesity in patients with T2D [16,28]. One could speculate that the elevated fat content in the OB mice could impair liver function and affect the secretion of MBL, or the reduction could be due to consumption, as suggested, for a reduction in circulating levels of C3 and C4 in patients with renal diseases [29].

By flow cytometry analysis, our in vitro study showed increased binding of MBL to human vascular endothelial cells cultured under hyperglycemic conditions relative to normoglycemic conditions, indicating an alteration in the presence of MBL binding sites, as previously suggested [18]. Elevated AGEs on endothelial cells in hyperglycemic conditions support this [30,31]. Additionally, endothelial progenitor cells (EPC) have been reported to be susceptible to apoptosis when cultured in a hyperglycemic environment, and transplantation of apoptosis-resistant EPC was suggested as a therapy treatment for diabetic kidney disease [32].

The increase in MBL-binding shown in our in vitro study was supported by the significantly higher protein levels of MBL-A and MBL-C found in kidney tissue and the isolated glomeruli from the OB mice as compared to the WT mice. Furthermore, immunofluorescence staining shows significantly increased binding of MBL-C in the glomeruli, together with increased C4 deposition in the OB mice. These observations are in line with our previous studies in a mouse model for T1D [17,18,19]. In the present study, we did not find evidence of MBL-A or MBL-C gene transcription locally in the glomeruli of either the OB or WT mice, which supports similar findings in MASP-2 knockout mice and WT mice [19,33].

In humans, increased levels of circulating MBL were reported to correlate with signs of diabetic kidney disease in both patients with T1D and T2D [16,34,35]. We have recently shown that patients with T2D and high circulating levels of MBL presented with impaired renal function [16]. Even though levels of MBL and C3 were higher in the circulation of the OB mice at 21 weeks, we did not find increased levels of C3 fragments in plasma. This may indicate a local, but not a systemic, complement activation. 

We observed increased adhesion of C4b in glomeruli in the OB mice as compared to the WT mice. Interestingly, the C4b seems to be localized with the involvement of the mesangial area, close to the macula densa region and the nearby situated tubular structure in the OB mice, whereas it was scattered all over the glomeruli in the WT mice. In support of our findings, increased C4 fragment deposition was found in T2DM patients with biopsy-proven diabetic kidney disease [36]. They found that C4c deposition in human glomeruli corresponded with the progression of the disease and was an independent risk factor for kidney dysfunction. Interestingly, we did not find a difference in C4 deposition in a previous study of a mouse model for T1D [19]. Of note, low-intense C4 deposition has been reported in glomeruli from non-diabetic WT mice and in normal human renal tissue [37,38].

In a recent systemic review of biomarkers in patients with T1D, Sarkar et al. reported that several circulating proteins related to the complement system, lipid metabolism, and immune response were upregulated [39]. They reported consumption of C4 and C3 via tissue deposition as an important marker of T1D. We and others have shown that the lectin pathway, followed by activation of the complement system, plays an important role in complications of T1D [6,7,8,13,40,41], the development of diabetic kidney disease [6,42], and high serum MBL levels are associated with the development of end-stage renal disease in patients with diabetic nephropathy [43].

The potent pro-inflammatory protein serum amyloid A 3 (SAA3), which is increased by the interaction between AGEs and the AGE-receptor (RAGE) to promote glomerular inflammation [44] and increased by chronic inflammation in both mice and humans [45], was measured. We found increased levels of SAA3 at both protein levels in plasma and renal gene expression in the OB mice. A link between complement activation and the NLRP3 inflammasome has been investigated in several studies, as previously reviewed [22,46]. Interestingly, we did not find a difference in the Nlrp3 or Asc gene expression, and caspase 1 showed a significantly lower expression in the OB mice as compared with the WT mice. Thus, although quantification of gene expression does not adequately measure the activity of the NLPR3 inflammasome, we did not find evidence of inflammation via this inflammasome in this mouse model for T2D and diabetic nephropathy from these analyses. We observed an increase in infiltrating CD68-positive cells in the glomeruli of the OB mice, in which we also discovered increased deposition and protein levels of MBL, which is in line with the observation by Ma et al. that overexpression of MBL in db/db mice promoted macrophage infiltration in the kidney [47]. Additionally, they reported increased expression of TNF-α in macrophages, but not in renal parenchymal cell lines, in culture after 12 h incubation with rMBL and suggest that MBL promotes renal fibrosis via crosstalk with mesangial and tubular epithelial cells [47]. Treatment with SGLT2 inhibitors in ob/ob mice showed improvement in the progression of diabetic nephropathy and was reported to occur in both a glucose-lowering-dependent and independent mechanism [48].

Due to a mutation in the leptin gene, the OB mice are heavily obese as compared to the WT BTBR mice, and fat was visually accumulated in all organs. The renal damage, which was confirmed by morphology, the UACR, and cystatin C levels, was not found to be caused by kidney fibrosis. We found increased levels of deposition of fibronectin in the glomeruli of the OB mice, but no difference was found in collagen deposition. A significant decrease was observed in the gene expression of collagen, the matrix metalloproteinase MMP-2, TIMP-2, and fibronectin in the kidney tissue, and no difference was observed in the expression of CTGF, MMP-9, TIMP1, or TGF-β; the latter has also been reported in male mice [49]. This supports the idea that the renal damage in this model is primarily a result of inflammation, which may progress to renal fibrosis. In support of the importance of inflammation, we found highly increased C3 mRNA and circulating C3 levels in the OB mice, which is likely to be regulated by increased TNF-α mRNA in the OB mice and supported by previous findings in rat glomerular endothelial cells [50].

## 4. Materials and Methods

### 4.1. Animals 

We used 4–5-week-old female BTBR OB mice (BTBR. Cg-Lep^ob^/WiscJ; Strain #004824) (OB) *n* = 30 and female BTBR wild-type (WT) non-diabetic controls (*n* = 30) (Jackson Laboratories, Bar Harbor, ME, USA), which is a well-established model for type 2 diabetes and diabetic nephropathy [21,51]. As OB mice exhibit increased mortality after 24 weeks of age [21], the mice in the present study were terminated after 21 weeks (Figure 11). The mice were housed in 3–7 per cage at a temperature of 21 ± 1 °C and humidity of 55 ± 5% with 12 h artificial light–dark cycles (light 6.00 a.m. to 6.00 p.m.). The mice had free access to water and chow (Lab Diet (5K52), as recommended by Jax.com). Eight animals were excluded from the study, all from the OB group; two animals were found dead in the cage during the study period, five were euthanized due to signs of infection, and one mouse did not develop diabetes. 

All animal experiments were performed under Danish regulations for the care and use of laboratory animals (license number 2017-15-0201-01330).

### 4.2. Study Design 

Animals were monitored daily for signs of illness and weekly during the whole study period for weight and blood glucose (Figure 10). Blood glucose was measured from tail vein blood by a contour glycometer (Bayer Diabetes Care, Kgs. Lyngby, Denmark). If blood glucose was above the limit of quantification (33.3 mmol/L), the missing value was replaced by 33.4 mmol/L in statistical evaluations. 

At week 19, the animals were placed individually in metabolic cages to collect urine. To diminish animal stress, including solitary containment without bedding, the animals were only held in isolation to collect urine for a timeframe of 16 h, and the 24 h urine production was estimated by extrapolation. At week 21, the animals were anesthetized by an intraperitoneal dose of ketamine at 0.5 mg/g and xylazine at 0.2 mg/g body weight (Ketaminol 4 Vet and Narcoxyl Vet, resp., Intervet, Skovlunde, Denmark). Both kidneys were removed, decapsulated, and weighed, and then the mice were sacrificed. From the right kidney, a small section of the kidney pole was snap-frozen for PCR analysis. The rest of the kidney was sagittally sectioned through the perihilar area. One half was fixed in 4% paraformaldehyde, and the other half underwent cryofixation embedded in Tissue-Tek (OCT compound, Sakura Finetek, Alphen aan den Rijn, The Netherlands). 

### 4.3. Blood Analysis

Blood samples were drawn twice during the study. Baseline samples were drawn from the mandibular vein puncture. The protocol stated baseline samples at week 6. However, due to the COVID-19 lockdown prohibiting access to the animal facilities, we were not able to collect the samples before week 12. As surrogate baseline samples, we purchased plasma samples from twelve 5-week-old mice, OB (*n* = 6) and WT (*n* = 6) (Jackson Laboratory). End-of-study samples were drawn by heart puncture after anesthetizing the animals. All samples were collected in 1.3 mL EDTA-coated tubes (Sarstedt, Nümbrecht, Germany). To minimize complement activation, the protease inhibitor Nafamostat mesylate (1 µL/100 µL blood, TCI EUROPE, Antwerp, Belgium) was added to the full blood samples immediately after collection. The samples were kept on wet ice until centrifugation at 4 °C and subsequently frozen and stored at −80 °C.

For the glycosylated hemoglobin (HbA1c) measurements, 50 µL full blood was collected in separate tubes and stored at −20 °C until analysis. HbA1c was measured on the Cobas C111 system (Roche Diagnostics, Rotkreuz, Switzerland). All procedures were conducted according to the manufacturer’s recommendations.

Biomarkers for kidney function, inflammation, and complement activation were measured by commercial ELISA kits: Mouse Cystatin C ELISA kit (DY1238, R&D Systems, Inc., Minneapolis, MN, USA), Mouse Serum Amyloid A3 ELISA (DY2948-05, R&D Systems), and Mouse Complement C3 ELISA kit (ab157711, Abcam, Cambridge, UK). All procedures were conducted according to the instructions.

Fragments of C3 (iC3b + C3dg) as a measure of complement activation were quantified by an in-house assay as previously described [52]. Samples were diluted by 1/50 and analyzed in duplicate.

Plasma MBL-A and MBL-C were measured with an in-house TRIFMA assay, as previously described [53]. In brief, MBL-A and MBL-C levels were captured in mannan-coated wells, followed by the addition of either biotinylated anti-MBL-A (Mab 3G6) or biotinylated anti-MBL-C (Mab 17E2) and europium-labeled streptavidin (PerkinElmer^®^, Hamburg, Germany). Samples were diluted at 1:2500 (MBL-A) and 1:5000 (MBL-C), respectively, and analyzed in duplicates. 

### 4.4. MBL-A and MBL-C Measurements in the Kidney Cortex and Glomerulus

The kidney cortex and glomerulus were homogenized in a phosphoinositide 3-kinase buffer and adjusted to a protein concentration in kidney tissue (8.44 µg/mL) and glomerulus (3 µg/mL). MBL-A and MBL-C concentrations in the kidney cortex and glomerulus were measured using the in-house TRIFMA protocol for plasma samples, as described above. 

For the analysis, the samples were measured in the following dilutions: MBL-A (1:100) and MBL-C (1:25). The anti-MBL-A and the anti-MBL-C antibody were diluted at 1:800 and 1:2680, respectively.

### 4.5. Urinalysis

The volume of the collected urine was measured and stored at −80 °C until further analysis. Urinary albumin concentration was measured by a Mouse Albumin ELISA Kit (Abcam, Ab108792), and urinary creatinine concentration was measured by a Creatinine Assay kit (Abcam, Ab204537). MBL-C was quantified as described above, with the exception that urine was diluted 1:2, and the results were adjusted for creatinine concentration. 

### 4.6. Histopathology and Immunohistochemistry

One-half of the harvested left kidney was fixated in 4% paraformaldehyde, embedded in paraffin, and cut into 5 μm tissue sections for histochemical staining. The observer was blinded by the group of animals. A Nikon Upright Microscope (lens ×40) (Nikon Instruments, Melville, NY, USA) and NIS element software were used for all immunohistochemical assessments. 

Mesangial morphology was characterized by Periodic Acid Schiff (PAS) staining. Mesangial index fraction was used to describe the glomerular damage, and measurements of glomerular hypertrophy were assessed in a minimum of 30 glomeruli per mouse (OB, *n* = 10, WT, *n* = 10). The PAS-positive mesangial index fraction was quantified as the percent fraction of the glomerular tuft using Weibel’s formel, as previously described [54]. 

### 4.7. Immunohistochemical Evaluation of Fibrosis Collagen and Fibronectin

Paraffin-embedded kidney sections (5 μm) were stained for collagen using Picrosirius Red (Abcam ab150681) according to the manufacturer’s instructions. Collagen staining in the Bowman’s capsule was visualized with the highly specific collagen-imaging method “linear polarized light detection” [55]. Collagen density within the Bowman’s capsule was estimated semi-quantitatively by assigning values on a 0–5 scale as follows: Grade 0; no color, grade 1; low-intensity color, grade 2; less than 50% with high-intensity color, grade 3; approximately 50% with high-intensity color, grade 4; above 50% with high-intensity color, grade 5; approximately 100% color of the bowman’s capsule. 

Kidney paraffin sections (5 μm) were deparaffinized, followed by heat-induced antigen retrieval in citrate buffer (10 mM, pH 6.0) (as described by Thermo Scientific, Waltham, MA, USA) in order to eliminate endogenous phosphatase activity. The sections were stained for fibronectin with a rabbit anti-mouse-fibronectin (Abcam ab2413) primary antibody and an alkalic phosphatase-conjugated secondary antibody (anti-rabbit AP-conjugated, Thermo 31346). StayRed/AP (Abcam ab103741) was used as a chromogen with an exposure time of 10 min. Sections without anti-mouse fibronectin antibodies were used as a negative control. Densities of fibronectin were quantified with an RGB color threshold measured by the percent fraction of the glomerular area. 

OCT-embedded kidney sections (8 μm) were blocked (Thermo R37629) and stained for macrophage infiltration of glomeruli. CD68-positive cells were evaluated by using a polyclonal rabbit anti-mouse CD68 antibody (Abcam, ab125212) in a dilution of 1:400 and a goat anti-rabbit IgG (H + L) Highly Cross-Adsorbed Secondary Antibody, HRP conjugated (Invitrogen, A16110, Waltham, MA, USA) in a dilution of 1:1000, with StayYellow/HRP as substrate chromogen (Abcam, ab169561) according to the manufacturer’s instructions. Twenty randomly chosen glomeruli per mouse (OB, *n* = 10, WT, *n* = 10) were evaluated, and all mononuclear cells with CD68+ reactions were counted.

The OCT-embedded kidney sections (8 μm) were fixed in acetone. The C4b and MBL-C staining were analyzed by: C4b: monoclonal rat anti-C4b antibody [16D2] (Abcam, ab11863) dilution 1:200; and MBL-C: rabbit anti-MBL-C antibody (mAb 14D12, Hycult Biotech, HM1038) dilution 1:20. The secondary antibodies used were a goat anti-rat-594 (Invitrogen, A11007) visualizing C4 and a goat anti-rabbit-488 (Invitrogen, A11008) visualizing MBL, both in a dilution of 1:500. The quantification of MBL-C and C4 was conducted by measuring fluorescence intensity within a region of interest (ROI), an area predefined as the glomerulus including the juxtaglomerular apparatus when visible (OB, *n* = 10, WT, *n* = 10). Nephrin: goat anti-mouse nephrin antibody (BioTechne, AF3159, Minneapolis, MN, USA) and nNOS: goat anti-mouse nNOS (GeneTex, GTX89962, Irvine, CA, USA), both dilution 1:50 and secondary, were donkey anti-goat-647 (ThermoFisher, A21447) diluted 1:500 and donkey anti-rabbit-488 (ThermoFisher, A21206) diluted 1:500 when in combination with C4 (OB, *n* = 5) and WT (*n* = 5). The images were processed by the software ImageJ (version 2.14.0/1.54f) [56].

### 4.8. Isolation of Glomeruli

The glomeruli were isolated, as described elsewhere [57]. In brief, the left kidney was cut into small pieces, and the tissue was placed in a digestion buffer (0.2% collagenase I and 7.5 μg/mL DNAse II) for 10 min at 37 °C. The tissue was then crushed with the end of a syringe and vigorously shaken for 10 min, followed by the addition of cold HBSS (Biowest, L0612, Nuaillé, France) to end the digestion. The cell suspension was passed through a 70 µm strainer (Corning^®^ CLS431751, Gilbert, AZ, USA), washed, and flow-through was collected and filtered through a 40 µm strainer (Corning^®^ CLS431750). The glomeruli captured in the 40 µm strainer were thoroughly washed 15 times in HBSS, snap-frozen in liquid nitrogen, and stored at −80 °C until use. 

### 4.9. Real-Time Quantitative PCR Analysis

Total RNA was isolated from snap-frozen kidney poles (OB, *n* = 22, WT, *n* = 30), primarily representing the cortex, using Trizol reagent (Invitrogen), and cDNA was synthesized using Verso™ cDNA synthesis kit 100RXN, AB-1453/B (VWR, West Chester, PA, USA). Real-time quantitative PCR was performed in duplicate with Lightcycler 480 SYBR Green I Master (Roche, Basel, Switzerland) using LightCycler^®^ 480 Instrument II (Roche). Data were normalized by the geometric mean of the following housekeeping genes: Ppia, Gapdh, and β-Akt. Finally, a melting curve analysis was performed. The increase in fluorescence was measured in real-time during the extension step, and the specificity of the amplification was checked by melting temperature analysis. Gene expression in the kidney with cycle threshold (Ct) values > 32 cycles was excluded from the result section. This was the case for mbl1 (MBL-A) and mbl2 (MBL-C). The data analysis was performed using the Livak method. The primer pairs are listed in Table 1 and were purchased from either LGC Biosearch Technologies, Lystrup, Denmark, or Eurofins Genomics, Ebersberg, Germany.

### 4.10. Flow Cytometry

For the analysis of change in the adhesion of MBL to endothelial cells in a hyperglycemic environment, confluent HUVECs were incubated in either 5.5 mM (+16 mM mannitol) or 21.5 mM glucose and harvested by trypsinization and counted for an amount of 35,000 live cells per tube (*n* = 8 per concentration, in 4 runs). The cells were incubated with rhMBL (10 µg/mL) and stained with FITC-conjugated anti-MBL antibody (5 µg/mL, HYP 131-1, Statens Serum Institut, Copenhagen, Denmark) FITC-conjugated with NHS-Fluorescein (Thermo) for 30 min at RT. TBS was used as a negative control. All samples were assayed on a NovoCyte Quanteon analyzer (Agilent Technologies, Santa Clara, CA, USA), and flow cytometry was performed at the FACS Core Facility, Aarhus University, Denmark. The MFI of a sample with 10 μg/mL of FITC-conjugated anti-MBL was divided by the MFI of a sample without added MBL to adjust for autofluorescence. 

### 4.11. Statistical Analysis

Data are presented as the mean (95% confidence interval (CI)) unless otherwise stated. Data distribution was evaluated using a QQ plot. Potential outliers were evaluated using the Rosner test. Normally distributed data were compared using the Student’s *t*-test, and non-normally distributed data were compared using the Wilcoxon Mann–Whitney rank sum test. A *p*-value below 5% was considered statistically significant. For the correlation analysis, Pearson was used when data were normally distributed, and Spearman was used when a non-parametric test was needed. Plots were computed as dot plots or boxplots with median, first, and third quartiles if non-parametric statistical testing or logarithmically transformed data were analyzed, or as mean and a 95% confidence interval if normal distributed data were analyzed. All analyses were carried out using RStudio version 2023.03.0 + 386.

## 5. Conclusions

We found increased staining of MBL-C and C4 in the glomeruli of the OB mice, as well as elevated inflammation by increased SAA3 in the circulation, higher TNF-α and C3 gene expression, and macrophage infiltration in the diabetic glomeruli. Furthermore, we discovered that MBL binds to human endothelial cells under hyperglycemic conditions, suggesting that our animal data are translatable to humans and that diabetes exhibits an altered cell surface pattern. 

## 6. Limitations

Because of a COVID-19 pandemic-ordered lockdown of the animal facility, we were not able to collect baseline samples until week 12. Therefore, the 5-week plasma samples were purchased from the Jackson Institute, and these mice can only refer to differences observed between the mice at week 5. This study only included female mice, and the results from the fibrosis analysis may have been different in male mice. 

## Figures and Tables

**Figure 1 ijms-25-07204-f001:**
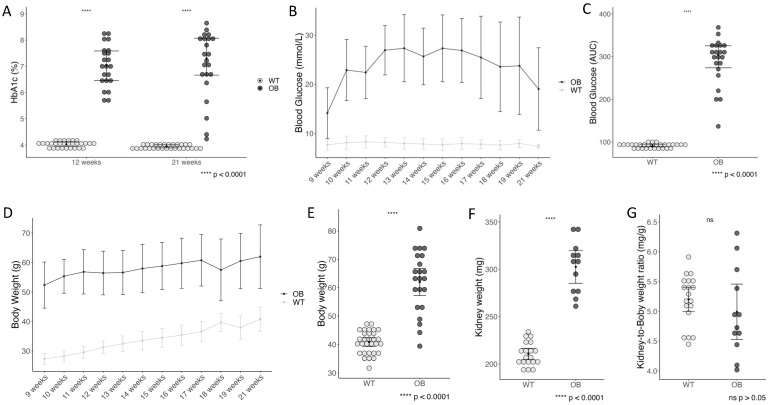
Diabetes status. (**A**) Level of glycosylated hemoglobin (HbA1c) at week 12 and week 21. (**B**) Development of the blood glucose levels. (**C**) Difference in area under the curve (AUC) of the blood glucose level. (**D**) Weight gain throughout the study period. (**E**) Difference in body weight at study end (week 21). (**F**) Kidney weight (WT *n* = 20, OB *n* = 14) and (**G**) kidney weight to body weight for the OB and WT mice. Data are presented as median and IQR, analyzed using the Wilcoxon Mann–Whitney rank sum test (**A**–**C**), presented as mean ± standard deviation (**D**), and mean with a 95% confidence interval, analyzed using the Student’s (**E**–**G**). Each mouse is represented by a dot (WT = gray (*n* = 30), OB = black (*n* = 22)).

**Figure 2 ijms-25-07204-f002:**
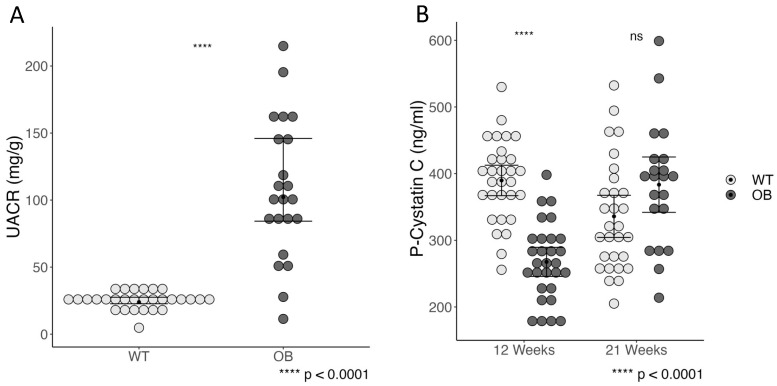
The 24 h urinary albumin excretion in the OB and the WT mice (**A**). Plasma cystatin C levels in OB and WT mice at week 12 and week 21 (**B**). Data are presented as the mean with a 95% confidence interval. Statistical analysis was performed using the Student’s *t*-test. Each dot represents a mouse: WT (gray, *n* = 30), OB (black, *n* = 22), ns = not significant (*p* > 0.05).

**Figure 3 ijms-25-07204-f003:**
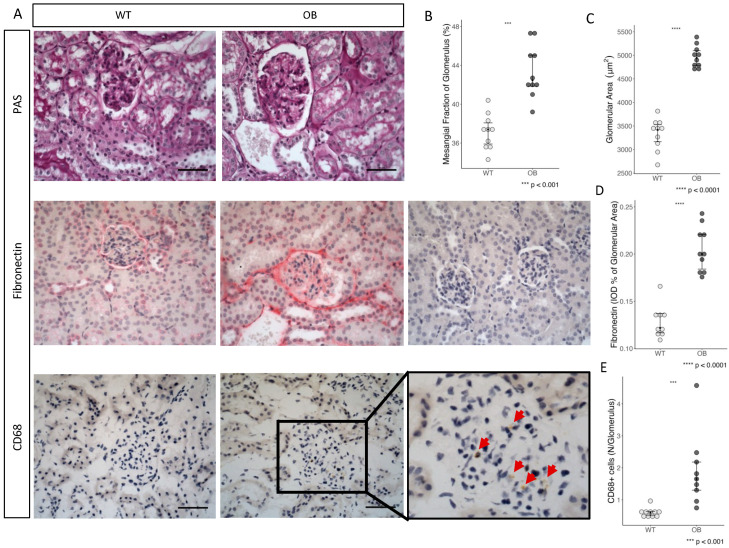
Renal histological examination and morphological changes. (**A**) Representative light microscopic pictures of PAS-staining, fibronectin-staining (without anti-fibronectin, represented at the far right as the negative control), and CD68-staining in the glomeruli of the OB and WT mice. All pictures are shown in ×40 lens, and the scale bar is 50 μm. In CD68 staining, only brown staining with coherent nucleus staining was included. (**B**) Estimation of mesangial area fraction as a percentage of glomerulus area. (**C**) Glomerular area. (**D**) Quantification of fibronectin-staining of glomeruli from OB and WT mice. (**E**) CD68+ cells per glomeruli. Red arrows indicate CD68+ cells. (**B**–**E**) Data are presented as the mean with the interquartile range (whiskers). Statistical analysis was performed using the Mann–Whitney U test, and each dot represents a mouse (*n* = 10 in each group).

**Figure 4 ijms-25-07204-f004:**
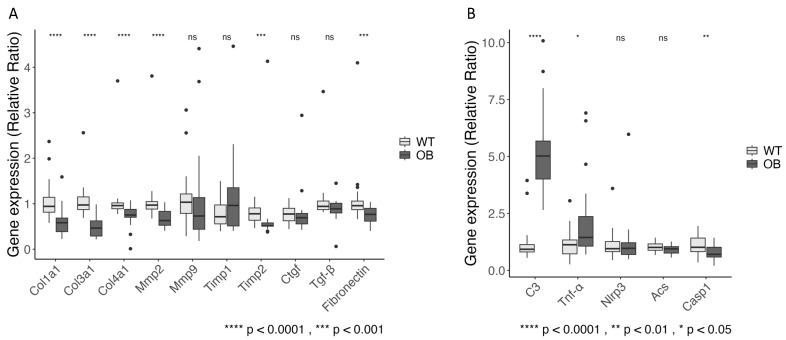
Renal cortex gene expression of targets in fibrosis development (**A**) and in inflammation (**B**). Data are presented as a relative ratio compared to WT, with the median (black line) and IQR shown in a boxplot. Statistical analysis was performed using the Wilcoxon Mann–Whitney rank sum test. (OB, black (*n* = 22), WT, gray (*n* = 30)). For Timp1, Timp2, Ctgf: OB, black (*n* = 18), WT, gray (*n* = 19), ns = not significant (*p* > 0.05).

**Figure 5 ijms-25-07204-f005:**
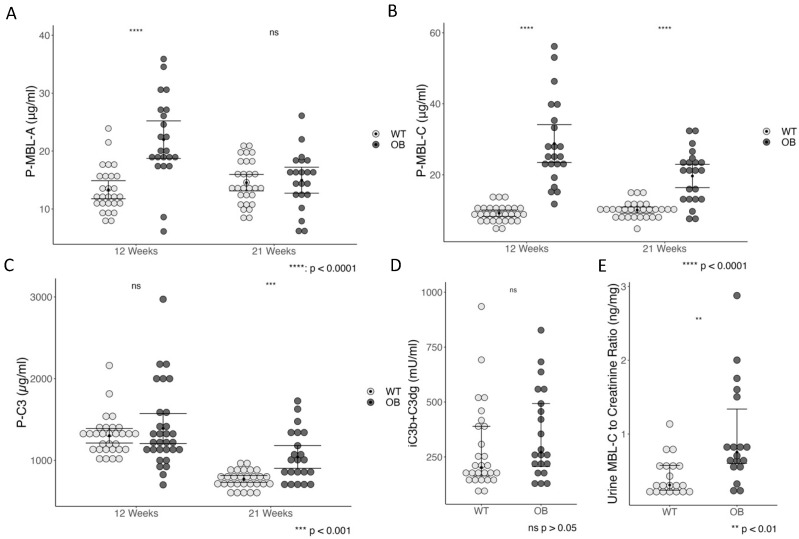
Circulating plasma levels of MBL-A (**A**) and MBL-C (**B**) in WT and OB mice. (**C**) Plasma levels of complement C3 in OB and WT mice. Data (**A**–**C**) are presented as the mean with a 95% confidence interval and analyzed using the Student’s *t*-test. (**D**) Activation of complement is determined by C3 fragments (iC3b + C3dg) plasma levels. (**E**) Concentration of MBL-C in urine. Data (**D**,**E**) are presented in median and IQR (whiskers) and analyzed using the Wilcoxon Mann–Whitney test. Each dot represents a mouse (WT (gray), *n* = 30; OB (black), *n* = 22), ns = not significant (*p* > 0.05).

**Figure 6 ijms-25-07204-f006:**
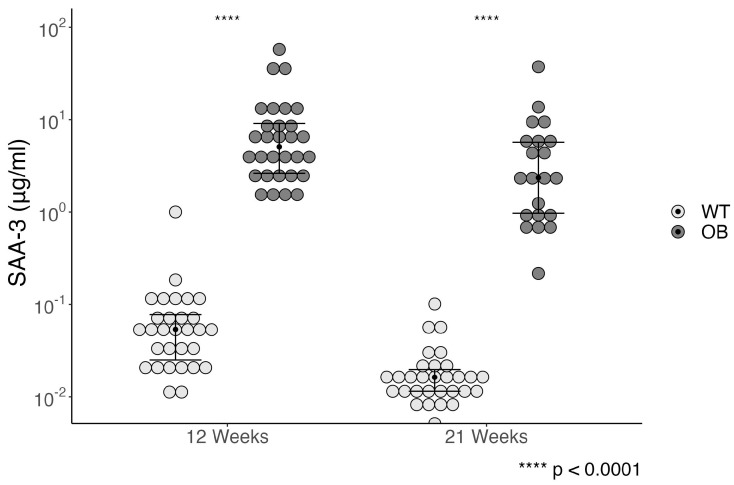
Systemic inflammation was assessed by Serum Amyloid A 3 (SAA-3) levels in OB (black, *n* = 22) and WT mice (gray, *n* = 30) at week 12 and week 21. Data are presented as median (black line) and IQR in a boxplot. Statistical analysis was performed using the Wilcoxon Mann–Whitney rank sum test. Data are presented with the *y*-axis on a logarithmic scale.

**Figure 7 ijms-25-07204-f007:**
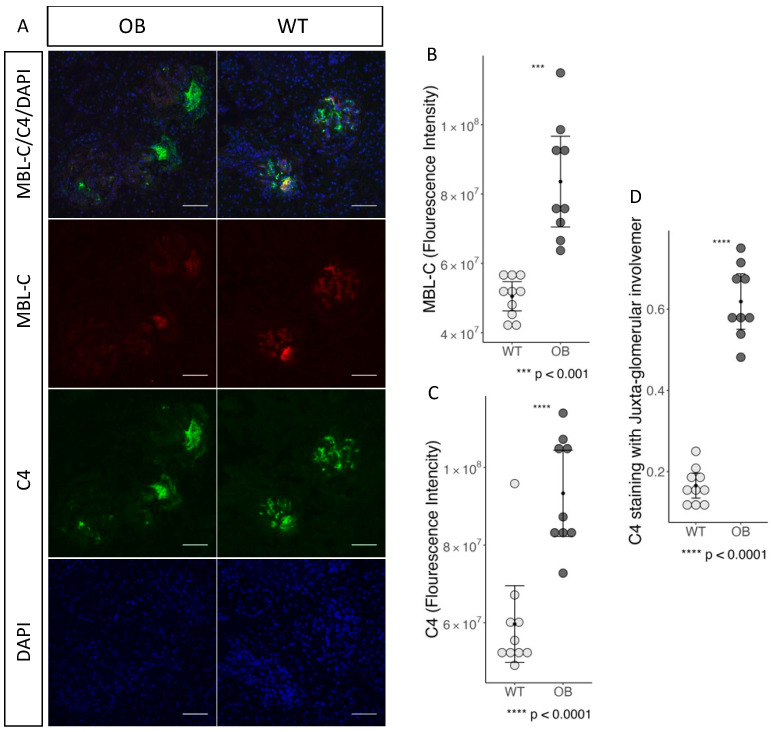
Complement activation. (**A**) Fluorescence immunohistochemical representation of glomerulus from OB (*n* = 10) and WT mice (*n* = 10). MBL-C (red), C4 (green), DAPI (blue), and a merged picture with all three wavelengths. ×40 lens is used, and the scale bar indicated in white is 50 μm. (**B**) MBL-C staining intensity within the region of interest selected as glomeruli. (**C**) C4 staining intensity within the region of interest selected as glomeruli. (**D**) Location of the C4 staining within the Juxtaglomerular region in the randomly selected glomeruli (*n* = 20 for each tissue section). All data are presented as the mean with a 95% confidence interval and analyzed by the Student’s *t*-test.

**Figure 8 ijms-25-07204-f008:**
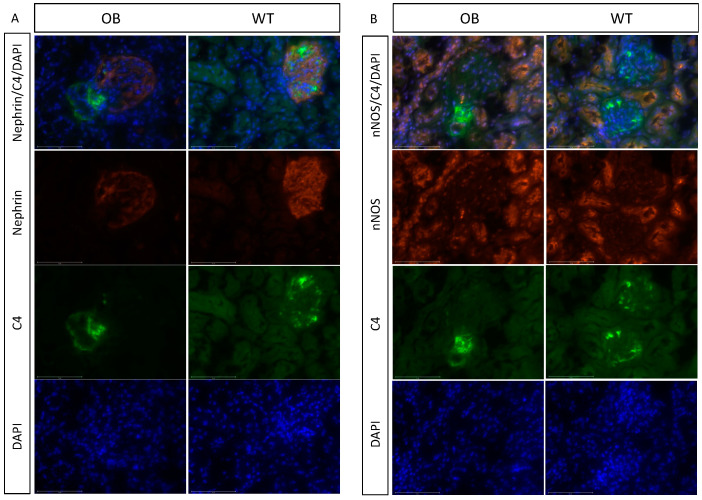
Deposition of C4 within the glomeruli. (**A**) Fluorescence immunohistochemical representation of C4 and nephrin in the glomerulus from OB (*n* = 5) and WT mice (*n* = 5). Nephrin (red), C4 (green), DAPI (blue), and a merged picture with all three wavelengths. EVOS with an x40 lens is used, and the scale bar indicated in white is 75 μm. (**B**) Immunohistochemical representation of C4 and nNOS. nNOS (red), C4 (green), DAPI (blue), and a merged picture with all three wavelengths. ×40 lens is used, and the scale bar indicated in white is 75 μm.

**Figure 9 ijms-25-07204-f009:**
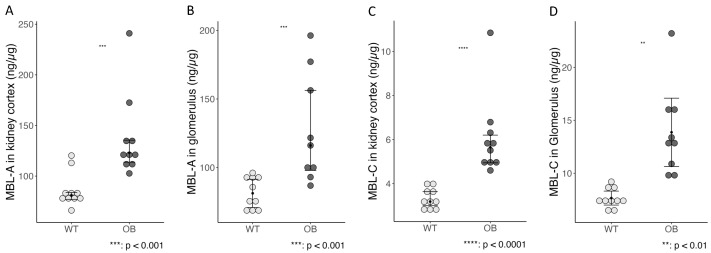
Protein levels of MBL-A and MBL-C were measured in the kidney cortex (**A**,**C**) and isolated glomeruli (**B**,**D**) of the OB and WT mice (OB, *n* = 10; WT, *n* = 10). Data are presented as mean with a 95% confidence interval (whiskers) and analyzed using the Student’s *t*-test (**A**–**C**) or as median (black dot) and IQR (whiskers) analyzed using the Wilcoxon Mann–Whitney rank sum test (**D**).

**Figure 10 ijms-25-07204-f010:**
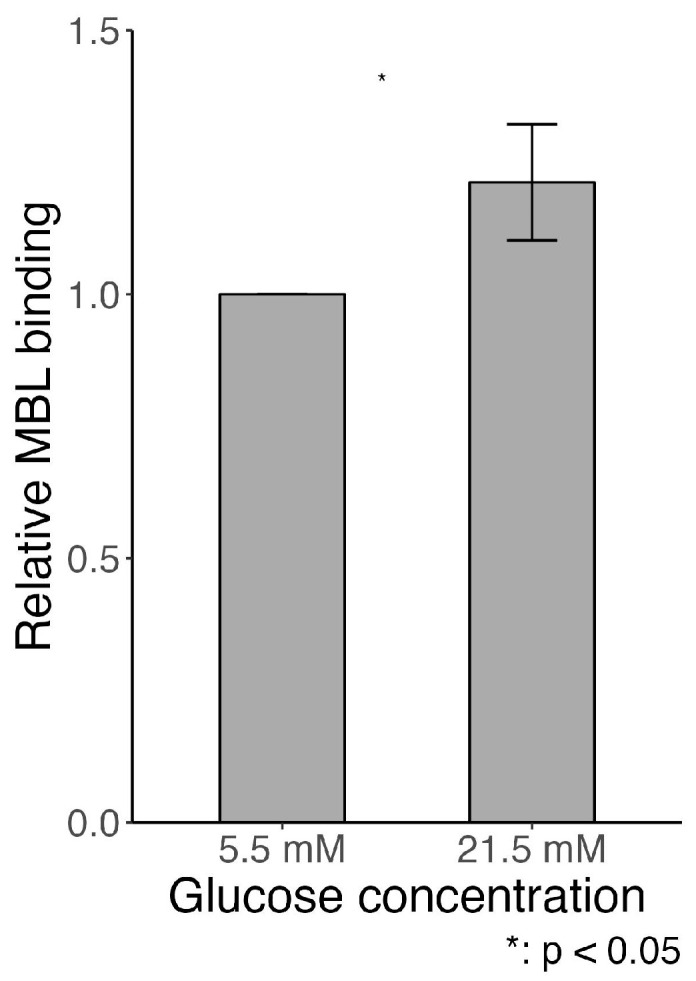
In vitro MBL binding to human vascular endothelial cells (HuVECs) cultured in either hyper- (21.5 mM glucose) or normo-glycemic (5.5 mM glucose + 16 mM mannitol) conditions for 15 days (+/−5 days). The cell study was repeated (*n* = 4), and each setup included eight replicates of each condition. Data presented as mean and ±standard deviation (whiskers) and analyzed using the Student’s *t*-test.

**Figure 11 ijms-25-07204-f011:**
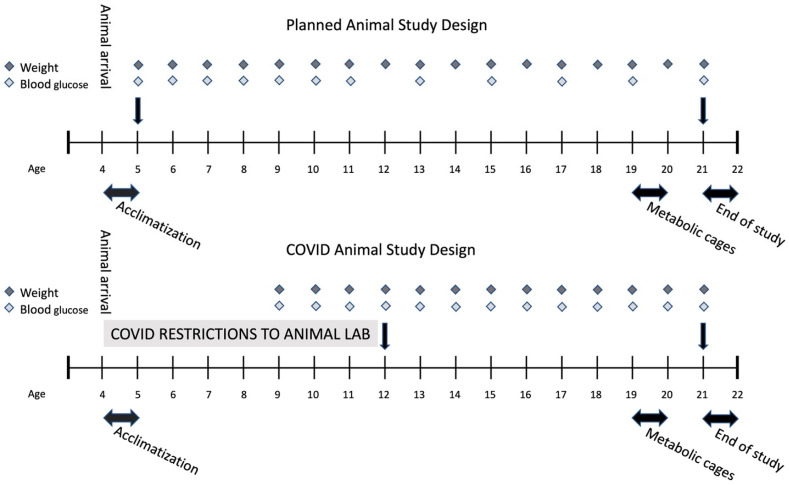
Overview of the study design and the alterations due to COVID-19 restrictions. Blood samples were indicated by the black arrow.

**Table 1 ijms-25-07204-t001:** Overview of forward and reverse primer sequences used to detect transcription of named genes.

Protein	Gene	Forward Primer Sequence	Reverse Primer Sequence
Collagen I	*Col1al*	GACGCATGGCCAAGAAGACA	ATTGCACGTCATCGCACACA
Collagen III	*Col3al*	GGGAATGGAGCAAGACAGTCTT	TGCGATATCTATGATGGGTAGTCTCA
Collagen IV	*Col4al*	ACTTCGCCTCCAGGAACGAC	GGTGCTTCACAAACCGCACA
Fibronectin	*Fn1*	ACATGCCTCGGGAATGGAAAGG	CGTCATAGCACGTTGCTTCATGG
Matrix Metalloproteinase 2	*Mmp2*	TCACTTTCCTGGGCAACAAGT	GCCACGAGGAATAGGCTATATCC
Matrix Metalloproteinase 9	*Mmp9*	TGAGTCCGGCAGACAATCCT	CGCCCTGGATCTCAGCAATA
Transforming growth factor-beta	*Tgfb1*	CTGACCCCCACTGATACGCC	GCGCTGAATCGAAAGCCCTG
Tumour necrosis factor-alpha	*Tnfa*	TTCCCAAATGGCCTCCCTCTCATC	TCCTCCACTTGGTGGTTTGCTAC
Mannan-binding lectin A	*Mbl1*	GTTCCCGGTCACCAGGCTAA	ACCACACACAGAAGGACAGGG
Mannan-binding lectin C	*Mbl2*	TGGGACCGAAAGGAGACCGT	AGGGCTCTCAGCTCTGATCGT
Complement factor 3	*C3*	CAGCTTCAGGGTCCCAGCTAC	CCAGCCGTAGGACATTGGGA
Caspase 1	*Casp1*	CACGCCCTGTTGGAAAGGAA	CCCTCAGGATCTTGTCAGCCA
Apoptosis-associated speck-like protein containing a CARD	*ASC*	AGAGTACAGCCAGAACAGGACA	CAGCACACTGCCATGCAAAG
NLR family pyrin domain containing 3	*Nlrp3*	GCTCCAACCATTCTCTGACCA	GGTTGGTTTTGAGCACAGAGG
Connective tissue growth factor	Ccn2	GGAGTGTGCACTGCCAAAGATG	AGGCAAGTGCATTGGTATTTGCAG
Tissue inhibitors of metalloproteinase 1	*Timp1*	GCCTACACCCCAGTCATGGA	GGCCCGTGATGAGAAACTCTT
Tissue inhibitors of metalloproteinase 2	*Timp2*	AGGAGATGTAGCAAGGGATCA	GAGCCTGAACCACAGGTACCA

## Data Availability

The raw data supporting the conclusions of this article will be made available by the authors upon request.

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
