# Peer review of "Mannan-Binding Lectin Is Associated with Inflammation and Kidney Damage in a Mouse Model of Type 2 Diabetes"

_ijms, 2024, doi:10.3390/ijms25137204_

Round 1

Reviewer 1 Report

Comments and Suggestions for Authors

The studies of Dorfinger and colleagues present the association of mannose-binding lectin with glomerular damage in female mice using a T2DM model. The mansucript is well written, the used methods are adequate and the results are convincing, although data presentation could be improved.

My comments are the following:

1) There is a rather high, 28% similarity with published articles, mostly found in figure legends and methods section. Please try to re-phrase those sentences, even if they are taken from the authors' previous publications.

2) The legend of Supplementary Table 1 is same as for regular Table 1, please correct.

3) Please provide a short explanation why female mice were used for the studies and not males. It is well known that female mice are more resistant to kidney damage in general.

4) For all figures: please provide exact sample sizes and description of used statistical methods for each figure.

5) Figure 2: UACR is not estimated, it is actually measured (from spot urine), pls revise figure legend.

6) What explains that plasma Cystatin-C was significantly lower in diabetic mice at 12 weeks?

7) Figure 3: Fibronectin IHC of the OB kidney clearly shows false positive intracapillary staining due to RBCs, also some interstitial and tubular cell cytoplasms are positive - I would modify the blocking step of the staining protocol. A larger magnification inset of CD68 stain should be presented, the actual images are not informative. Why only n=10 kidneys were evaluated when experimental sample size was much larger? Pictures obviously show a x400 magnification, not x40 as stated.

8) To determine fibrosis, the authors should measure Timp1 and Timp2 mRNA expression along with Mmp2 and Mmp9, the latter alone is not informative about matrix remodeling. Besides, Timp1 has been recently shown to be a crucial player in kidney damage (doi: 10.1016/j.kint.2022.03.029). How would you explain that Col1 and Col3 expression decreased, whereas Fn1 expression increased in diabetic mice? Tgfb1 mRNA itself is not enough when investigating fibrosis (as Tgfb1 mRNA and protein expression does not parallel most of the time)  - please check TGFB1 protein level (eg. immunostaining), and Ctgf mRNA expresion.

9) Figure 4 and Figure 6: Why box and whiskers was used here? Dot plots should be used throughout the whole paper to show individual values. Also, sample size and statistics is missing.

10) Figure 7: How many kidneys were stained? There are 9-10 dots per group in each graph, does it mean only 9-10 kidneys were stained from the 22/30 kidneys available? Magnification can not be x40 either.  From these fluorescent photos it is not obvious we see a glomerular staining. A nephrin or synaptopodin additional staining is recommended.

11) Figure 9 legend says the cells were treated with 21.5 mM glucose, text in methods and results states 25mM glucose was used. Which one is true? In this figure, bar chart should be changed to dot plots.

12) Please delete lines 72-75, that remained from MDPI template.

13) Lines 330-331 in discussion: This sentence can be debated, as inflammation leads to fibrosis, especially in the glomerulus. Inflammatory cytokines induce mesangial matrix production and cell proliferation, as well as podocyte damage. As proteinuria was detected, some parameters showing podocyte integrity are needed (eg nephrin and podocin mRNA expression, nephrin IHC). 

14) In Table 1, please show murine genes in italics.

15) Apart of MBL binding, why did you not investigate other aspects of HUVEC cells in the experiment? Eg. hyperglycemia induced pro-fibrotic gene expressions, etc.

Author Response

Dear Editor,

We appreciate the constructive and thoughtful comments you and the reviewers provided and are grateful for the opportunity to send you a revised manuscript. We hope that we have addressed your concerns satisfactorily but remain open to further revision.

Thank you for your attention to our paper.

Sincerely yours,

Mette Bjerre

Please find our point-by-point responses to the reviewers' comments. All modifications are highlighted in red in the manuscript.

Reviewer 1

The studies of Dorflinger and colleagues present the association of mannose-binding lectin with glomerular damage in female mice using a T2DM model. The manuscript is well written, the used methods are adequate and the results are convincing, although data presentation could be improved.

Reply: We thank the Reviewer for the efforts in reviewing our manuscript and for the positive and constructive comments. We have done our best to address the concerns below.

My comments are the following:

1) There is a rather high, 28% similarity with published articles, mostly found in figure legends and methods section. Please try to re-phrase those sentences, even if they are taken from the authors' previous publications.

Reply: Similarities within the method section and figure legend may be difficult to completely avoid. We have rephrased part of the method section and included additional information to the method section and the figure legends as suggested.

2) The legend of Supplementary Table 1 is same as for regular Table 1, please correct.

Reply: A legend for the supplementary Table 1 is now included (Line 530): Biomarker of the lectin pathway, nephropathy, and inflammation in 5-week-old mice.

3) Please provide a short explanation why female mice were used for the studies and not males. It is well known that female mice are more resistant to kidney damage in general.

Reply: Thank you for the comment. We agree that there is a sex difference in the model regarding renal changes in diabetes. However, with reference to Hudkins et al. JASN 2010, we believe these differences in this particular diabetes model are modest. In the planning of the study, weighing pros and cons between mouse sex, including these phenotypical characteristics and practical considerations, including differing housing requirements of female mice vs. male mice we ultimately chose to use female mice. As mentioned under limitations, only female mice are included in our study, and the results may differ from those of male mice.

4) For all figures: please provide exact sample sizes and description of used statistical methods for each figure.

Reply: We have included the sample size and statistical methods in each figure legend.

5) Figure 2: UACR is not estimated, it is actually measured (from spot urine), pls revise figure legend.

Reply: Thank you for the opportunity to correct this. We have corrected the legend for Figure 3.

6) What explains that plasma Cystatin-C was significantly lower in diabetic mice at 12 weeks?

Reply: Thank you for this comment. Cystatin C is an early marker of renal damage and the level of Cystatin C increases when kidney function and the GFR decreases. The fact that plasma cystatin C levels are lower in the OB mice at 12 weeks could be a sign of glomerular hyperfiltration as a compensatory mechanism.

7) Figure 3: Fibronectin IHC of the OB kidney clearly shows false positive intracapillary staining due to RBCs, also some interstitial and tubular cell cytoplasms are positive - I would modify the blocking step of the staining protocol. A larger magnification inset of CD68 stain should be presented, the actual images are not informative. Why only n=10 kidneys were evaluated when experimental sample size was much larger? Pictures obviously show a x400 magnification, not x40 as stated.

Reply: Thank you for the opportunity to clarify this. The paraffin sections were boiled in citrate buffer to eliminate endogenous phosphatase activity. This information is now included in the method section (line 433): Kidney paraffin sections (5 μm) were deparaffinized followed by heat induced antigen retrieval in citrate buffer (10 mM, pH 6.0) (as described by Thermo scientific) in order to eliminate endogenous phosphatase activity. Additionally, we have included a negative control/without primary stained section for fibronectin together with new representative staining Figure 3.

We have included a larger magnification inset for the CD68 stain, as suggested in Figure 3. All tissue stainings were performed on a subgroup including 10 mice from each group, as mentioned in the legend for Figure 3, and now clearly stated in the methods section.

The lens used for the immunohistochemistry is x40, which is now clarified in the figure legend of Figure 3 and Figure 7 and in the method section. As the images were evaluated on a computer screen and not on prints of fixed size, the final magnification varies between screens. To accurately present size on the images, scalebars representing 50 mm are included on the presented photos.

8) To determine fibrosis, the authors should measure Timp1 and Timp2 mRNA expression along with Mmp2 and Mmp9, the latter alone is not informative about matrix remodeling. Besides, Timp1 has been recently shown to be a crucial player in kidney damage (doi: 10.1016/j.kint.2022.03.029). How would you explain that Col1 and Col3 expression decreased, whereas Fn1 expression increased in diabetic mice? Tgfb1 mRNA itself is not enough when investigating fibrosis (as Tgfb1 mRNA and protein expression does not parallel most of the time)  - please check TGFB1 protein level (eg. immunostaining), and Ctgf mRNA expresion.

Reply:  As shown in Figure 4, all collagens analyzed (col1, col3, col4) and fibronectin (Fn1) decreased in the renal tissue of OB mice compared with WT mice. The condition of the kidneys (and livers) from the OB mice were highly infiltrated with fat, which may lower the mRNA transcript of the ECM genes. As inflammation precedes the development of fibrosis, the timing of the tissue extraction may affect the results. An explanation may be the investigation of female mice, in which fibrosis may develop differently as compared to male mice or at a later time point. Similar TGFb expression in renal tissue from BTBR WT and ob/ob mice has been reported by Opazo-Rios et al. 2022, which is now included in the discussion (line 337)

9) Figure 4 and Figure 6: Why box and whiskers was used here? Dot plots should be used throughout the whole paper to show individual values. Also, sample size and statistics is missing.

Reply: Box plot were used for the gene expression (Figure 4), which we believe is a general form to present these data. We have changed the box plot in Figure 6 to dot plots as suggested and included sample size and statistical methods in both figure legends.

10) Figure 7: How many kidneys were stained? There are 9-10 dots per group in each graph, does it mean only 9-10 kidneys were stained from the 22/30 kidneys available? Magnification can not be x40 either.  From these fluorescent photos it is not obvious we see a glomerular staining. A nephrin or synaptopodin additional staining is recommended.

Reply: Thank you for the possibility to clarify this. We have included 10 mice from each group for the histopathology and immunohistochemistry evaluation. This information is now included in the method section and the number of mice is included in the legends. The lens used for the immunohistochemistry is x40, which is now clarified in the figure legend and the scalebar represented in the pictures is 50 mm as mentioned. We have improved the quality of the pictures for better visualization of the glomeruli.

11) Figure 9 legend says the cells were treated with 21.5 mM glucose, text in methods and results states 25mM glucose was used. Which one is true? In this figure, bar chart should be changed to dot plots.

Reply: Thank you for this observation. The cells were incubated with 21.5 mM glucose, as shown in Figure 9. The glucose concentration is now corrected in the methods section 4.10 line 495.

12) Please delete lines 72-75, that remained from MDPI template.

Reply: Lines 72-75 are now deleted.

13) Lines 330-331 in discussion: This sentence can be debated, as inflammation leads to fibrosis, especially in the glomerulus. Inflammatory cytokines induce mesangial matrix production and cell proliferation, as well as podocyte damage. As proteinuria was detected, some parameters showing podocyte integrity are needed (eg nephrin and podocin mRNA expression, nephrin IHC). 

Reply: Thank you for this comment and for the opportunity to elaborate on this. We agree that chronic inflammation, seen in obesity and T2D, leads to development of fibrosis. We have rephrased the sentence (lines 334-335). “This supports that the renal damage in this model is primarily a result of inflammation which may progress to renal fibrosis.”

14) In Table 1, please show murine genes in italics.

Reply: The genes in Table 1 are now shown in italics.

15) Apart of MBL binding, why did you not investigate other aspects of HUVEC cells in the experiment? Eg. hyperglycemia induced pro-fibrotic gene expressions, etc.

Reply: Thank you for this comment. The purpose of the in vitro study was to support our in vivo findings. Our focus of the in vitro study was to investigate if human endothelial cells exposed to hyperglycemia (mimicking the high blood glucose levels in patients with diabetes) would generate increased potential binding sites for MBL. This was indeed the case, and the in vitro results support that alterations of the EC cell surfaces are caused by hyperglycemia. The fibrotic and inflammatory genes investigated in the mice may not be transferable to the in vitro system, since the settings are markedly different in terms of complexity and a single cell type.  

Reviewer 2 Report

Comments and Suggestions for Authors

Current manuscript, authors aimed to determine the role of Mannan-biniding lectin (MBL) complement factor in mouse model of diabetic nephropathy. Authors used animal model of kidney disease associated with T2D, female BTBR OB mice. This mouse model is well established as kidney diseases associated with glomerular injury, sclerosis associated with inflammatory mediated pathways involved including role of macrophages (M1/M2) in progression of kidney disease. However, in the current manuscript, authors found increased MBL in circulation and within the glomeruli of OB/OB mice together with increased complement C3 levels. This study appears, very descriptive and lack of novelty and indicating results cannot determine the casual relationship with disease progression, and did not address the role of MBL in worsening the disease progression. All the indicated results (includes biochemical parameters, inflammatory markers and histology) have been established in previous studies including the activation of complement system. The current study design lacks the mechanistic experiments that can suggest blockage or decrease or inhibit the accumulation of MBL associated with ameliorate the kidney disease progression.  Authors did not clearly indicate why female mice has been used in this study.  

Author Response

Reviewer 2

Current manuscript, authors aimed to determine the role of Mannan-biniding lectin (MBL) complement factor in mouse model of diabetic nephropathy. Authors used animal model of kidney disease associated with T2D, female BTBR OB mice. This mouse model is well established as kidney diseases associated with glomerular injury, sclerosis associated with inflammatory mediated pathways involved including role of macrophages (M1/M2) in progression of kidney disease. However, in the current manuscript, authors found increased MBL in circulation and within the glomeruli of OB/OB mice together with increased complement C3 levels. This study appears, very descriptive and lack of novelty and indicating results cannot determine the casual relationship with disease progression and did not address the role of MBL in worsening the disease progression. All the indicated results (includes biochemical parameters, inflammatory markers and histology) have been established in previous studies including the activation of complement system. The current study design lacks the mechanistic experiments that can suggest blockage or decrease or inhibit the accumulation of MBL associated with ameliorate the kidney disease progression.  Authors did not clearly indicate why female mice has been used in this study.  

Reply: We thank the Reviewer for the efforts in reviewing our manuscript. We have done our best to address the concerns.

We and others have shown that MBL plays a role in T1D diabetes using different STZ diabetes induced models. In clinical studies, elevated circulating MBL levels correlate with the progression of diabetic complications, including renal disease; however, low MBL levels have also been associated with increased cardiovascular mortality. The novelty in our in vivo study, investigating the ob/ob mice, a model of T2D and nephropathy, shows elevated MBL levels in the circulation, which are captured in the renal tissue and especially in the purified glomeruli as a consequence of diabetic nephropathy. Additionally, our in vitro study shows increased MBL binding to human endothelial cells cultured under hyperglycemic conditions, supporting/confirming the hypothesis that hyperglycemia alters self-cells for recognition by the innate immune system. In the discussion, we have included a more detailed description of the role of MBL in worsening the disease progression (marked in red). 

As described above, the sex difference in the model regarding renal changes in diabetes was considered before the selection of mice. With reference to Hudkins et al. JASN 2010, we believe these differences in this particular diabetes-model are modest. In the planning of the study, weighing pros and cons between mouse sex, including these phenotypical characteristics and practical considerations, including differing housing requirements of female mice vs. male mice, we ultimately chose to use female mice.

Reviewer 3 Report

Comments and Suggestions for Authors

In the present article, the authors investigate the role of some pathophysiological factors, that lead to diabetic nephropathy in the mouse model of type 2 diabetes, isolated glomerular cells, and endothelial cells (HUVECs) in a hyperglycemic environment. The authors have used a wide methods range. The statistical analysis is adequately selected, and the presented results and discussion correlate with a short conclusion. 

I have the following recommendations for authors:

The reference list should be corrected. It needs to be added citations from the last five years. 

Comments on the Quality of English Language

Minor grammatical errors were found, which may be corrected.

Author Response

Reviewer 3

In the present article, the authors investigate the role of some pathophysiological factors, that lead to diabetic nephropathy in the mouse model of type 2 diabetes, isolated glomerular cells, and endothelial cells (HUVECs) in a hyperglycemic environment. The authors have used a wide methods range. The statistical analysis is adequately selected, and the presented results and discussion correlate with a short conclusion. 

Reply: We thank the Reviewer for the efforts in reviewing our manuscript and for the positive and constructive comments.

I have the following recommendations for authors:

The reference list should be corrected. It needs to be added citations from the last five years. 

Reply: The reference list has been corrected and we have included additional references as suggested by the reviewer when suitable.

Cai et al. Mannose-binding lectin activation associated with the progression of diabetic nephropathy in type 2 diabetes mellitus patients. Ann Transl Med 2020 Nov;8(21):1399. doi: 10.21037/atm-20-1073.

Hudkins et al Regression of diabetic nephropathy by treatment with empagliflozin in BTBR ob/ob mice. Nephrol Dial Transplant 2022 Apr 25;37(5):847-859.doi: 10.1093/ndt/gfab330.

Opazo-Rios et al Kidney microRNA expression pattern in Type 2 Diabetic Nephropathy in BTBR Ob/Ob mice. Front. Pharmacol 2022 Volume 13 https://doi.org/10.3389/fphar.2022.778776

 Minor grammatical errors were found, which may be corrected.

 Reply: We have corrected grammatical errors.

Round 2

Reviewer 1 Report

Comments and Suggestions for Authors

The authors answered to most of my comments and revised the manuscript accoringly.

However, the authors did not provide the asked Timp1, Timp2 and Ctgf gene expression results, or the TGF-beta protein expression data, that are very easy to measure and would substantially improve the unclear fibrosis-related chapter. Without TGFB1 protein data, Tgfb1 mRNA expression results are useless, given from the nature of post-translational TGF-beta maturation process. To some extent, Ctgf gene expression would overcome this issue as it has direct impact on fibrosis. Similar problem exists with MMP2 and MMP9 without corresponding TIMPs, as I already pointed out in my previous review. The balance of ECM production and degradation relies mostly on MMP/TIMP interplay in the kidney, therefore these parameters need to be investigated and discussed together, picking only MMPs does not make sense. So, either include Timp1 and Timp2 or delete both Mmp results.

The authors have also omitted my previous comment 10 about glomerular immunostaining by providing additional nephrin/synaptopodin staining for podocytes or vimentin for mesangial cells. In the present Figure 7, there is no information which cells are positively stained, although in the WT kidneys the C4 distribution pattern looks like mesangium. As the stainings in OB kidneys show a very distorted pattern, further approach is needed to identify the cells involved in MBL and C4 immunoreactivity. For JGA region, you can stain macula densa cells for nNOS, for instance.

Another question rose here, as WT glomeruli should normally not present any complement immunoreactivity, which is a clear sign of autoimmune glomerulonephritis. This issue need to be discussed in detail!

Author Response

Dear Editor,

We hope that we have addressed the reviewers remaining concerns satisfactorily, but we remain open to further revision if needed.

Please note that we will not be able to send the revised manuscript with the 10 days period for resubmission as additional analysis are requested by reviewer 1.

We aim to send the revised manuscript no later than June 21th 2024, as we have to extract new RNA for the gene expression analysis and order additional antibodies for the immunostainings.

Mette Bjerre

Reviewer 1

The authors answered to most of my comments and revised the manuscript accordingly.

Reply: We thank the Reviewer for accepting the majority of the revised manuscript. We have done our best to address the last concerns below.

However, the authors did not provide the asked Timp1, Timp2 and Ctgf gene expression results, or the TGF-beta protein expression data, that are very easy to measure and would substantially improve the unclear fibrosis-related chapter. Without TGFB1 protein data, Tgfb1 mRNA expression results are useless, given from the nature of post-translational TGF-beta maturation process. To some extent, Ctgf gene expression would overcome this issue as it has direct impact on fibrosis. Similar problem exists with MMP2 and MMP9 without corresponding TIMPs, as I already pointed out in my previous review. The balance of ECM production and degradation relies mostly on MMP/TIMP interplay in the kidney, therefore these parameters need to be investigated and discussed together, picking only MMPs does not make sense. So, either include Timp1 and Timp2 or delete both Mmp results.

Reply: We will include analysis of timp1, timp2 expression and as suggested the ctgf gene expression will be performed instead of the TGFB-1 immunostaining. However, due to lack of material, we will not be able to include all the mice in this analysis. The actual number of mice will be noted in the legend for Figure 4. New extractions for the gene expression analysis are needed and the revision will thus exceed the 10 days for submission of the revised manuscript.

The authors have also omitted my previous comment 10 about glomerular immunostaining by providing additional nephrin/synaptopodin staining for podocytes or vimentin for mesangial cells. In the present Figure 7, there is no information which cells are positively stained, although in the WT kidneys the C4 distribution pattern looks like mesangium. As the stainings in OB kidneys show a very distorted pattern, further approach is needed to identify the cells involved in MBL and C4 immunoreactivity. For JGA region, you can stain macula densa cells for nNOS, for instance.

Reply: It was not our intention to omit question 10 in the first revision. We understood this as a need for improvements of the quality of pictures. Therefore, we revised Figure 7 for an improved visualization of MBL and C4 location in the glomeruli. However, we will now include staining with nephrin and nNOS as suggested by the reviewer. This will require an extension of the time needed for the submission of the revised manuscript.

Another question rose here, as WT glomeruli should normally not present any complement immunoreactivity, which is a clear sign of autoimmune glomerulonephritis. This issue need to be discussed in detail!

Reply: Thank you for this comment. Previous studies have also shown positive staining for C4 in glomeruli of non-diabetic mice, however no difference in staining intensity was observed (C4 and C3) Ostergaard et al 2016 [19] and low intense C4 staining is reported in a non-specific mesangial pattern in control mice (Luo et al 2018, doi: 10.3389/fimmu.2018.01433), which is also reported in normal human renal tissue (Zwirner et al 1989, Kidney international vol 36 (1989) 1069-1077). This is now included in the discussion.

Lines 293-300: We observed increased adhesion of C4b in glomeruli in the OB mice as compared to the WT mice. Interestingly, the C4b seems to be localized around the juxtaglomerular area in the OB mice, whereas it was scattered all over the glomeruli in the WT mice. In support of our findings, increased C4 fragment deposition was found in T2DM patients with biopsy-proven diabetic kidney disease [36]. They found that C4c deposition in human glomeruli corresponded with the progression of the disease and was an independent risk factor for kidney dysfunction. Interestingly, we did not find a difference in C4 deposition in a previous study of a mouse model for T1D [19]. Of note, low intense C4 deposition has been reported in glomeruli from WT mice and in normal human renal tissue (Luo et al, and Zwirner et al).

Reviewer 2 Report

Comments and Suggestions for Authors

Authors claimed that the novelty in our in vivo study, investigating the ob/ob mice, a model of T2D and nephropathy, shows elevated MBL levels in the circulation, which are captured in the renal tissue and especially in the purified glomeruli as a consequence of diabetic nephropathy.  However, there is no evidence for causal relationship between increased circulating MBL with diseases progression and current study design lacks the mechanistic experiments that can suggest blockage or decrease or inhibit the accumulation of MBL associated with ameliorate the kidney disease progression. 

Author Response

Reviewer 2

Authors claimed that the novelty in our in vivo study, investigating the ob/ob mice, a model of T2D and nephropathy, shows elevated MBL levels in the circulation, which are captured in the renal tissue and especially in the purified glomeruli as a consequence of diabetic nephropathy.  However, there is no evidence for causal relationship between increased circulating MBL with diseases progression and current study design lacks the mechanistic experiments that can suggest blockage or decrease or inhibit the accumulation of MBL associated with ameliorate the kidney disease progression. 

Reply: As mentioned in the first revision note, this study was designed to investigate the involvement of MBL in diabetic nephropathy in a model of T2D. We have reported importance of MBL in this model with increased binding of MBL in glomeruli both in renal sections and in purified glomeruli. We do agree that investigations of blockage or inhibition of MBL or complement is an interesting path to pursue, however, this is beyond the scope of this manuscript.

Round 3

Reviewer 1 Report

Comments and Suggestions for Authors

The new measurements of mRNA expression and further immunostainings will improve the manuscript, we are waiting for these results.

In the meantime, please revise the statistical analysis in Figure 3, as histology evaluation should be analyzed by non-parametric tests, such as Mann-Whitney. Student's t-test is incorrect here.

Author Response

Comment: The new measurements of mRNA expression and further immunostainings will improve the manuscript, we are waiting for these results.

Reply: We thank the Reviewer for accepting our revision proposals. We have revised the manuscript accordingly.

mRNA expression: We have included the gene expression analysis of timp1, and timp2 as requested and added the ctgf gene expression instead of the TGFB-1 immunostaining. Due to lack of material, it was not possible to include all the mice in this analysis. The actual number of mice for the additional three analysis has been noted in the legend for Figure 4 (Timp1, Timp2, Ctgf: OB, black (n=18), WT, gray (n=19)). The primers are included in Table 1

We found no significant difference in the expression of Timp1 and Ctgf (which was also the case for TGF-b) between OB and WT mice, but a decrease in expression of Timp-2 was found in the OB mice. Some studies (reviewed by J. Clin. Med. 2020, 9, 472; doi:10.3390/jcm9020472) have found early up-regulation of mmp-2 and -9 but down-regulation of both mmps after a longer exposure to hyperglycemia, which may explain our findings. We find a tendency of increased TIMP-1 expression in the OB mice, however, this is not significant. The importance of TIMP-1 in diabetic kidney disease seems to differ according to the mouse genetic backgrounds (Kidney Int. 2022 Aug; 102(2): 337–354.  doi: 10.1016/j.kint.2022.03.029), that may explain the lack of difference in the BTBR. Cg-Lepob/WiscJ mice used in our study.

Our new data support the overall conclusion of the manuscript, that the renal damage in this model is primarily a result of inflammation, which may progress to renal fibrosis.

Immunostaining: We have included staining with nephrin and nNOS in combination with C4 as suggested by the reviewer. The procedure and antibodies used are described in the method section line 492: Nephrin: goat anti mouse nephrin antibody (BioTechne, AF3159) and nNOS: goat anti mouse nNOS (GeneTex, GTX89962) both dilution 1:50 and secondary were donkey anti goat-647 (ThermoFisher, A21447) diluted 1:500 and in donkey anti rabbit-488 (ThermoFisher, A21206) diluted 1:500 when in combination with C4 (OB, n=5), WT (n=5).

Comment: The results are shown in Figure 8 (new figure added to the revised manuscript, and the following figures are re-numbered).

Figure 8. Deposition of C4 within the glomeruli. A) Fluorescence immunohistochemical representation of C4 and Nephrin in glomerulus from OB (n=5) and WT mice (n=5). Nephrin (red), C4 (green), DAPI (blue), and a merged picture with all three wavelengths. EVOs with x40 lens is used and the scale bar indicates in white is 75 mm. B) Immunohistochemical representation of C4 and nNOS. nNOS (red), C4 (green), DAPI (blue), and a merged picture with all three wavelengths. x40 lens is used and the scale bar indicates in white is 75 mm.

Our additional data support the presence of C4 in the JGA described in Figure 7D. The localization of C4 staining in the glomeruli was evaluated by podocyte staining using Nephrin (Figure 8A). The increased C4 deposition in OB mice did not seem to be situated in the proximity of the podocytes. However, staining by nNOS, visualizing the tubular region and the macula densa region, showed a potential involvement of the mesangial area, close to the macula densa region and the nearby situated tubular structure (Figure 8B). 

Discussion line 317: We observed increased adhesion of C4b in glomeruli in the OB mice as compared to the WT mice. Interestingly, the C4b seems to be localized with involvement of the mesangial area, close to the macula densa region and the nearby situated tubular structure in the OB mice, whereas it was scattered all over the glomeruli in the WT mice.

Comment: In the meantime, please revise the statistical analysis in Figure 3, as histology evaluation should be analyzed by non-parametric tests, such as Mann-Whitney. Student's t-test is incorrect here.

Reply: Thank you for this comment. The plots and statistics have been corrected in Figure 3.

Reviewer 2 Report

Comments and Suggestions for Authors

Thank you. The authors addressed to the comments . 

Author Response

Comment 1: Thank you. The authors addressed to the comments . 

Reply: Thank you